# The Cultural Heritage of “Black Stones” (*Lapis Aequipondus*/*Martyrum*) of Leopardi’s Child Home (Recanati, Italy)

**DOI:** 10.3390/ma15113828

**Published:** 2022-05-27

**Authors:** Patrizia Santi, Stefano Pagnotta, Vincenzo Palleschi, Maria Perla Colombini, Alberto Renzulli

**Affiliations:** 1Dipartimento di Scienze Pure e Applicate, Università degli Studi di Urbino Carlo Bo, Campus Scientifico “Enrico Mattei”, Via Cà Le Suore 2, 61029 Urbino, Italy; patrizia.santi@uniurb.it; 2Dipartimento di Scienze della Terra, Università di Pisa, Via Santa Maria 53, 56126 Pisa, Italy; stefano.pagnotta@unipi.it; 3Applied Laser Spectroscopy Laboratory, ICCOM-CNR U.O.S., Pisa, Via G. Moruzzi 1, 56124 Pisa, Italy; vincenzo.palleschi@cnr.it; 4Dipartimento di Chimica e Chimica Industriale, Università di Pisa, Via G. Moruzzi 13, 56124 Pisa, Italy; maria.perla.colombini@unipi.it

**Keywords:** ultramafic metamorphic rock, stone artefact, *Lapis Aequipondus*/*Martyrum*, cultural heritage, counterweight, LIBS, XRF

## Abstract

A macroscopic lithological study and physical (hardness, size, weight) investigations, coupled with laser-induced breakdown spectroscopy (LIBS) and X-ray fluorescence (XRF) chemical analyses of three egg- and one pear-shaped polished black stones, exposed in the library of the child home of the famous poet Giacomo Leopardi, at Recanati (Italy), were carried out. They are characterized by different sizes: two with the same weight of 16.9 kg and the two smaller ones of 5.6 kg each, corresponding to multiples of standard roman weights (*drachma* and *scrupulum*). These features and the presence of some grooves on the rock artefacts, probably for grappling hooks, suggest an original use as counterweight for the four black stones herein classified as amphibole-bearing serpentinites whose lithologies are far away from Recanati (probably coming from geological outcrops in Tuscany). The four serpentinite stones closely match with the so-called *Lapis Aequipondus* used in antiquity by the Romans as counterweights. Due to the presence of lead rings or iron hooks in these stones, *Lapis Aequipondus* were also used for martyrdoms during the persecution of Christians in the Roman period, attached to the necks of martyrs that were then thrown in the wells or attached to the ankles of hanging bodies. This is the reason why these stones are also known as *Lapis Martyrum*, venerated with the relative martyrs, in several churches of Rome. The four black stones investigated probably arrived at Recanati from Rome after the middle of the 19th century. In the past, Christians also called *Lapis Martyrum* the “devil’s stones” (*Lapis Diaboli*). This could also be the reason for the popular belief that black stones cannot be touched by people, except those of the Leopardi dynasty. This work contributes to the cultural heritage of Leopardi’s child home, as the four black stones had never been investigated.

## 1. Introduction

In the child home of the famous Italian poet Giacomo Leopardi (born in Recanati on 29 June 1799 and died in Naples on 14 June 1837), some black stones are exposed on the table of one of the library’s rooms. Recanati is a small village (at present, of about twenty thousand inhabitants) in central Italy (Marche Region) just 12 km away from the Adriatic Sea.

Historical data on the provenance of these stones and the way they arrived to Recanati are not available. Most probably, they reached Recanati after the middle of the 19th century, since, according to Count Vanni Leopardi, one of the last members of the Leopardi family at the time of this study (he later died in 2019), they are not recorded in the detailed catalog of rare objects written by Monaldo Leopardi (1776–1847), father of the famous poet. During the shooting of the movie *Il giovane favoloso* (Italy, 2014; director, Mario Martone), dedicated to the life of Giacomo Leopardi, the actor Massimo Popolizio (Monaldo Leopardi in the movie) was impressed by the fact that the director said to him that, according to a popular belief, “only people of the Leopardi family could touch the black stones” [1], as other people touching them would be hit by a curse. 

That probably may derive from the mysterious atmosphere around these stones because of their exotic appearance and the black color, which is very different from the chromatic features (pale-yellow to whitish) of the sedimentary rocks in the surroundings of Recanati [2]. Some historical reasons at the base of the popular idiom should, however, exist, and they could be unraveled through the present work on these stone artefacts based on a petrographic macroscopic observation (naked eye and hand lens), morphological and physical (size, weights, hardness) studies and a chemical survey, conducted through laser-induced breakdown spectroscopy (LIBS) and X-ray fluorescence (XRF). As a matter of fact, this is the first petrographic investigation on the four black stones exposed in the library of the child home of the famous poet Giacomo Leopardi. Qualitative results of the analyses (LIBS and XRF spectra) will be discussed and compared in order (i) to confirm that the combination of these two techniques is useful for chemical investigations in the field of cultural heritage and (ii) to determine the petrographic classification of the stones. In addition, petrographic and physical comparisons of the four black stones with similar lithologies known and used in antiquity will be carried out and addressed in an archaeometric framework. In particular, of paramount importance will be a comparison, by lithology, of physical properties and/or shape with (a) similar black stones, which are present nowadays in several churches in Rome [3,4,5,6,7], (b) the available geological information of similar rocks, which are present in limited outcrops in Tuscany and used as building or ornamental stones in this Italian region [8,9,10] and (c) photographic collections of similar black stones [11,12]. Unfortunately, no analyses with the LIBS and XRF were possible to carry out in similar black stones found in the Roman churches (no permission was given). However, comparisons based on robust mineralogic, chemical and petrographic methods enabled an archaeometric study of the four black stones, leading to reasonable conclusions concerning their significance in the framework of the cultural heritage of Leopardi’s child home.

## 2. Materials and Methods

The investigated stones consist of three roughly egg-shaped and one pear-shaped polished rock artefacts (Figure 1). With permission of Count Vanni Leopardi, a series of physical measurements (size, weight) and macroscopic observation of physical properties (color, shining, hardness) of the rock type were carried out. No sampling was allowed, but in situ chemical analyses of the four stones were, nevertheless, permitted. The analysis was performed using mobile dual-pulse instrument for laser-induced breakdown spectroscopy (Modì), a LIBS system made by Marwan Technology (Pisa) and Elio, a portable X-ray fluorescence (XRF) instrument by Bruker Co. These techniques require no sample pre-treatment, a considerable advantage with respect to traditional spectroscopic destructive techniques, which need sample mineralization by acid attack. LIBS and XRF are somewhat complementary elementary techniques. The XRF technique is very versatile, but it cannot detect elements lighter than Al (atomic weight 13); it provides volume-integrated analysis and has a lateral spatial resolution, typically of the order of 1 mm. On the other hand, the LIBS technique can detect light elements, such as Na (atomic weight 11) or Mg (atomic weight 12), has a lateral spatial resolution of the order of a few tens of micrometers and provides an in-depth compositional analysis [13]. The identification and the assignment of emission lines relevant to single atomic species allows us to determine the sample elemental composition. The calibration-free LIBS method was used for the quantitative analysis of the LIBS spectra (CF-LIBS) [14]. In fact, CF-LIBS does not require calibration standards, since all the information needed for the determination of the sample composition is extracted from the LIBS spectrum itself. The main drawback of LIBS, on the other hand, is its micro-destructivity. The joint use of XRF and LIBS thus allows us to exploit the benefits of both techniques, mitigating their drawbacks (for example, reducing the number of LIBS points of analysis by performing a preliminary compositional screening using the non-destructive XRF technique). These techniques were additionally coupled with fundamental lithological analyses and physical properties characterization. In addition, the comparison with compatible stone artefacts (by lithology) used in antiquity (throughout Italy) and their relative use/s was performed, thus allowing us to focus the present study on the four black stones in the framework of a contribution to the cultural heritage of Leopardi’s child home as well. Table 1 synthesizes the methods and rationale adopted in the present study.

## 3. Results

### 3.1. Physical Properties, Lithology, Mineralogy and Petrography

The four black stones are similar from a macroscopic point of view and can thus be all classified as belonging to the same lithology, a fine- to medium-grained metamorphic rock consisting of mafic minerals. No orientation of minerals occurs (isotropic structure). The rocks cannot be scratched with the fingernail and hardly scratched with a sharpener or a coin. In this way, they are characterized by a medium-high hardness, between 5 and 7 in the Mohs scale. In addition, they are partially greasy to the touch.

On the rock surfaces, traces of partially smoothed and circular grooves (up to 1 cm wide) are present (in the upper portions of stone 1, 2, 4 of Figure 1). The grooves may have been originally used to fix a harness or a ring and the relative hook (grappling hooks?). Two out of four rock artefacts are partially damaged in the upper portion, probably due to their use, where the mechanical stress of the harness (or ring) produced a rock rupture (upper portions of stone 3 and 4; Figure 1). Although the black color prevails in the well-polished surfaces, having a metallic shine, millimetric to centimetric pale- to olive-green spots are also present, mostly where some roughness occurs or along the grooves. As shown in Figure 1, the stones have different sizes: two (1 and 4) with the same weight of 16.9 kg and two smaller ones (2 and 3) of 5.6 kg each. These weights roughly correspond to five Roman *drachmae* (i.e., 17.04 kg; 1 Roman *drachma* = 3.408 kg) and five Roman *scrupuli* (i.e., 5.68 kg; 1 Roman *scrupulum* = 1.136 kg). The two largest (16.9 kg) egg-shaped stones have the two axes of the rotation ellipsoid constituting a solid ovoidal form of ca. 23.4 × 36.4 and 24.4 × 41.3 cm, respectively, whereas the smaller egg-shaped one has the two axes ca. 18.7 × 26.7 cm. The pear-shaped stone has a maximum width of ca. 36.3 cm. 

The main mafic minerals recognized macroscopically (naked eye and hand lens) are serpentine and amphibole. The LIBS and XRF spectra (Figure 2) are very similar for each of the four samples and reveal a chemical qualitative composition, which is referrable to a serpentinite. In fact, from the LIBS spectrum, a stoichiometric ratio Mg/Si between 1.4 and 1.6 is evaluated using the calibration-free LIBS (CF-LIBS) approach developed by ALS Lab [14], which agrees with the serpentine formula Mg_6_[(OH)_8_Si_4_O_10_]. The presence of clear peaks of Ca and Fe (both in LIBS and XRF spectra) indicates the presence of amphibole of the tremolite-actinolite species Ca_2_(Mg,Fe)_5_[OH,F(Si_4_O_11_)]_2_. In addition, the presence of peaks of Na and Al (LIBS) suggests the possible presence of low amount of jadeite (a pyroxene, NaAlSi_2_O_6_). As the four black stones are ultramafic metamorphic rocks of the greenschist facies [15], deriving from the serpentinization of mantle peridotites, it is not surprising to also see in the spectra Ni (XRF) and Cr (both LIBS and XRF) peaks, which are probably from relict minerals of olivine and Cr-spinel. The sulphur peak (XRF) agrees with the common presence of sulphides in serpentinites as accessory minerals [16], possibly pyrite (FeS_2_) or chalcopyrite (CuFeS_2_), as Cu also appears in both LIBS and XRF spectra. Traces of sphalerite [(Zn,Fe)S] can also be inferred due to Zn peak in the XRF spectrum. As the XRF analysis reported in Figure 2 was performed close to the smoothed grooves originally used to fix a harness or a ring and the relative hook, the detected traces of Pb could be referrable to grappling hooks sealed with lead (plumbing). Weak peaks of strontium, visible in both XRF and LIBS spectra, could be associated with Ca having a chemical affinity (charge, ionic radius) and usually present in accessory minerals, such as calcite (CaCO_3_), in this kind of serpentinite lithology. The XRF spectrum shows the characteristic fluorescence emission of the X-ray tube anode (Rh), while the emission lines from ambient air (N, O, H) are detectable in the LIBS spectrum. 

### 3.2. Comparative Results with Black Serpentinites Used in Antiquity

In the four black stones investigated, the correspondence of the weights to multiples of some standard Roman weights and the presence of grooves to hook a harness or a ring clearly address their origin as counterweights. The lithological similarity to what is already known as *Lapis Aequipondus* or *Lapis Martyrum* [4,5,17] is straightforward (Figure 3). The Romans utilized this stone to make scale weights. There were weights of different sizes, and for this reason, lead rings or iron hooks were attached to them. According to Corsi [17], *Lapis Aequipondus* or *Lapis Nephriticus* has a relatively high hardness, pale to olive green color, belonging to the jade group of rocks. Jade is used as a synonym of Na-pyroxenite, having Na-pyroxene (jadeite, Fe-jadeite, Mg- and Fe-omphacite or a mix of them) as the most abundant phase. It corresponds only in part to the gemological term «jade» [18,19], in which nephrite is also included [20]. The term “*Nephriticus*” clearly refers to the nephrite, which is an amphibole of the tremolite-actinolite series, Ca_2_(Mg, Fe)_5_Si_8_O_22_(OH)_2_ whose name was, however, abolished by the International Mineralogical Association (IMA). In addition, a misunderstanding may exist concerning the *Lapis Aequipondus*, as the ancient Roman stone makers also gave the same name to a dark green to black serpentinite, partially greasy to the touch, that can be well polished and again used for counterweights [17]. The four black stones of Leopardi’s child home thus clearly refer to this latter variety of *Lapis Aequipondus*.

In the framework of mafic and ultramafic metamorphic rocks of the greenschist facies used in antiquity and greasy to the touch, the gray to pale green soapstones (talc- and magnesite-bearing chlorite schist) recognized as a subgroup of the so called pietra ollare [21,22,23,24] can be ruled out (for different modal mineralogy, color and hardness) as the lithology of the four stones of Leopardi’s child home.

The rings or hooks of the *Lapis Aequipondus* also allowed the use of these stones as martyrdom instruments during the Christian persecution [3,5,6,17], mostly attaching these stones to the neck of the martyrs who were then thrown in wells. *Lapis Aequipondus* is, therefore, generally known as *Lapis Martyrum* and also as the “devil’s stone” (*Lapis Diaboli*) [6]. The use of *Lapis Martyrum* is well represented in some frescos of the Basilica of Santo Stefano Rotondo (Rome; Figure 4), with the stone (a spherical or flattened ball shape) attached to the ankles of hanging bodies to make the tortures more painful. This is the reason why these stones started to be highly venerated by the Christians as *Lapis Martyrum*, and several ovoidal to disc-shaped or truncated cone samples (e.g., with a shape of a circular loaf) are present nowadays in several Roman churches, such as Santa Maria in Cosmedin, Santa Maria in Trastevere, Santa Sabina and San Lorenzo Fuori le Mura (Figure 5).

Examples of *Lapis Martyrum* are reported in the literature [5,7] in other Roman churches, such as S. Clemente, S. Prassede, S. Pudenziana, S. Paolo alle Tre Fontane, S. Nicola al Carcere Tulliano, SS. Cosma and Damiano, museums (Musei Capitolini and Museo Lateranense of Rome) and S. Angelo church of Perugia. Finally, a series of weight units for measurement are exposed at the Museum of Terme di Diocleziano (Rome).

## 4. Discussion and Conclusions

The black stones found at Giacomo Leopardi’s child home had never been classified according to mineralogy and petrography. As no destructive analyses were permitted on the artefacts’ material, LIBS and portable XRF chemical surveys therefore represented fundamental analytical techniques, which allowed, along with macroscopic structure investigation and physical properties analyses, to define (for the first time) an appropriate and rigorous petrographic classification of the four black stones that arouse so much curiosity at Leopardi’s child home: amphibole-bearing serpentinites. Comparisons with stone artefacts possessing very similar lithologies and physical features also lead to a recognition of the four black stones as counterweights, known by the Romans as *Lapis Aequipondus* and also used for martyrdoms (*Lapis Martyrum*) during the persecutions against Christians.

However, such metamorphic rocks are not present in the surroundings of Recanati, nor within the radius of hundreds of kilometers from the hometown of the famous poet, an area only characterized by sedimentary rocks. As a matter of fact, serpentinites (greenschist facies) have their origin in rocks belonging to ophiolite sequences and, according to the knowledge on the regional petrology and geological maps of Italy, the best candidate areas for their provenance can be represented by Liguria and Tuscany regions in the Northern Apennines [25,26]. Of course, ophiolite sequences are also widespread in the Alps and throughout the Mediterranean area, but the presence of some old quarries of green to dark green to black serpentinites exploited in ultramafic metamorphic rocks of the ophiolite sequences of Tuscany and Liguria addresses our lithological comparisons to central Italy. As a matter of fact, various lithotypes of serpentinites from the Northern Apennines were largely used in the historical architecture of the above two regions [27]. The area of Pian di Gello near Monte Ferrato, north of Prato, was recognized by Del Riccio and Rodolico [3,10] as a source area of black serpentinites used for building stones. In particular, in his “History of the Stones”, the monk Agostino Del Riccio [3] describes a “Paragone” Stone near Prato (Sacca and Sant’Anna), which can be considered another variety of dark green to black serpentinite. He also reports the term “Frombole di Mare” for some of these black (“Paragone”) polished stones with rounded shape, which were present in Rome and used for Christian persecutions during the Roman period [3]. 

A dark green to black serpentinite quarried in the surroundings of Florence (near Antella) was used to pave the floor of Santa Maria del Fiore Cathedral in Florence [8]. By contrast, from the Sacca di Prato quarry come the black “Paragone” stone and the green “Green of Prato” stone used for the external walls of the Cathedral itself [8]. Finally, it is worth noting that Giamello et al. [9] indicate dark green to black serpentinite quarried in the area of Casciano of Murlo/Vallerano (Siena) as the source area for many lithotypes used on the floor of the Siena Cathedral. 

Clues therefore exist to locate the most probable area of provenance of the four black stones of Leopardi’s child home (and thus of *Lapis Aequipondus* or *Lapis Martyrum*), namely, the quarrying sites of Tuscany. Although Monaldo Leopardi (the father of the famous poet) used to record every kind of object present in the home, nevertheless, there is not a trace of them in the Leopardi archive. It can thus be hypothesized that the four black stones probably arrived in Recanati, from Rome, after the death of Monaldo (after the mid-19th century). As the Leopardi family was deeply Catholic, the bad reputation of the four black stones and the legend about their curse on all people touching them (except those of the Leopardi dynasty) could probably derive from the use of this kind of stone as *Lapis Martyrum*. In any case, the overall study of these stones, from petrographic classification to comparison with other artefacts and their use in the past, sheds light on a matter stimulating curiosity to all the visitors of Leopardi’s child home, where the four black stones are exposed in a room of the library. 

## Figures and Tables

**Figure 1 materials-15-03828-f001:**
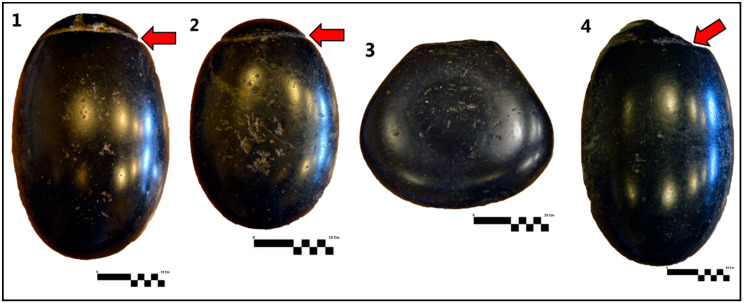
The four black stones. Two with the same weight of 16.9 kg (**1**,**4**) and two smaller ones (**2**,**3**) of 5.6 kg each. The red arrows indicate the traces of partially smoothed grooves originally used to fix a harness or a ring and the relative hook (or a grappling hook). The graphic scale is 10 cm (the small black and white squares in the right part are 1 cm^2^ each). ©Famiglia Leopardi Recanati. Courtesy of Leopardi family, any reproduction is forbidden.

**Figure 2 materials-15-03828-f002:**
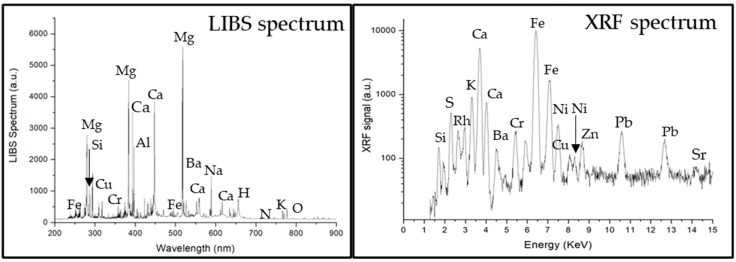
Representative analyses of the four investigated black stones by laser-induced breakdown spectroscopy (LIBS) and X-ray fluorescence (XRF). Spectra (both LIBS and XRF) are very similar for each of the four black stones of the Casa Leopardi’s child home.

**Figure 3 materials-15-03828-f003:**
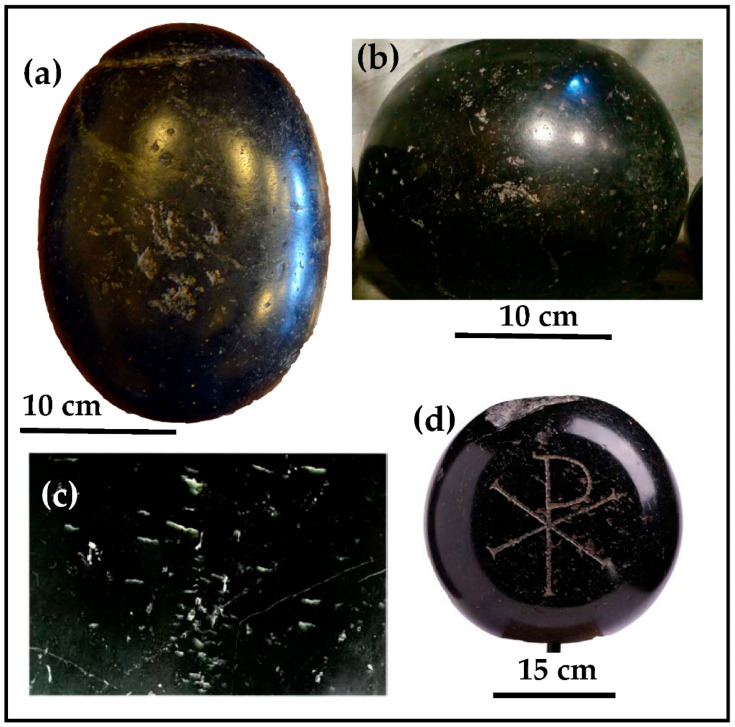
Macroscopic comparison among: (**a**) one of the black stones (1) from Leopardi’s child home; (**b**) *Lapis Martyrum* located at Santa Maria in Trastevere church (Rome); (**c**) particular of *Lapis Aequipondus* with permission of G. Giardini who took the photo [11] at the Lateran Museum (Rome); (**d**) *Lapis Martyrum* 4th century AD (private collection [12]).

**Figure 4 materials-15-03828-f004:**
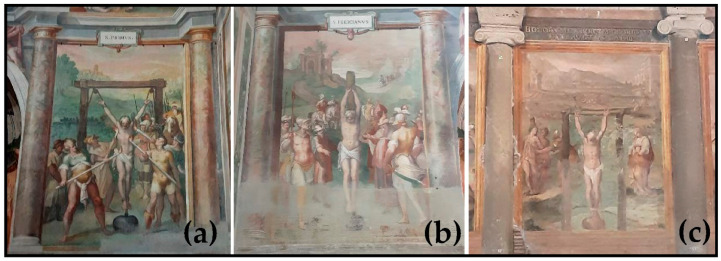
Indoor frescos inside the Basilica of Santo Stefano Rotondo, Rome (Antonio Tempesta, Pomarancio and Matteo da Siena, 1586) representing the martyrdom of: *San Primus* (**a**); *San Felicianus* (**b**); a nameless martyr (**c**). *Primus* and *Felicianus* brothers suffered martyrdom around the year 297 AD, during the Diocletian persecution. The relatively bad quality of photos (**b**,**c**) is because the frescos are not well preserved.

**Figure 5 materials-15-03828-f005:**
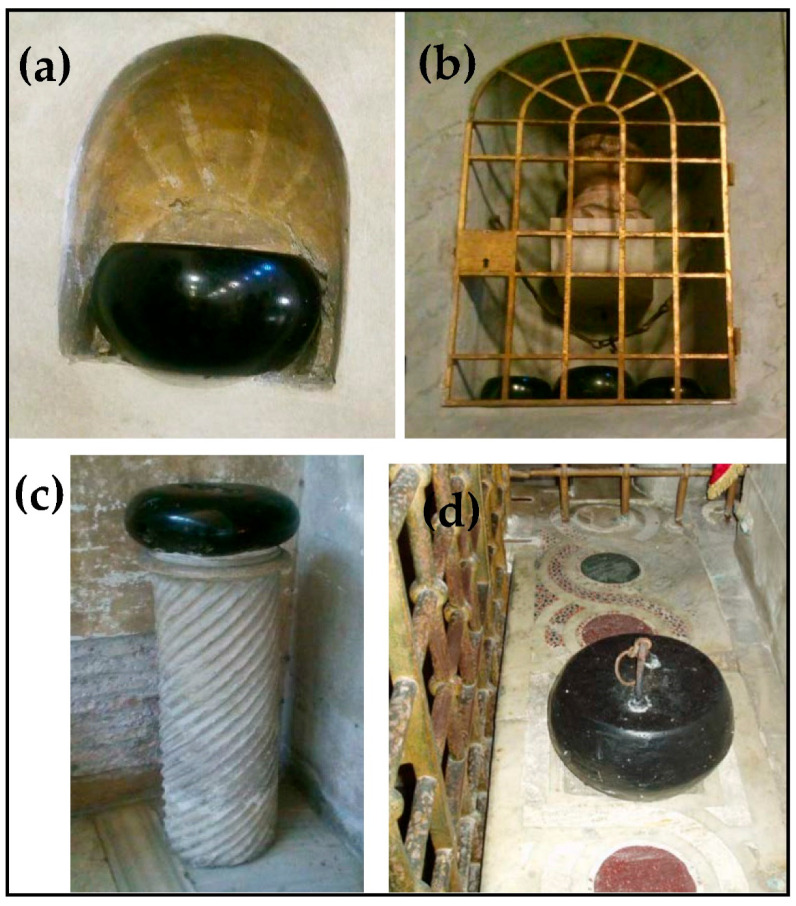
Some *Lapis Martyrum* hosted in different churches of Rome; (**a**) Santa Maria in Cosmedin; (**b**) Santa Maria in Trastevere; (**c**) Santa Sabina; (**d**) San Lorenzo Fuori le Mura (presently missing).

**Table 1 materials-15-03828-t001:** Methods and rationale of the present study (in four phases) aimed at unraveling the origin of the investigated black stones of Leopardi’s child home.

1	Analysis of the four black stones of the Leopardi’s child home (LIBS, XRF, structure, lithology, colour, shining, hardness)	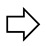	Petrographic classification
2	Comparison with the stone artifacts used in antiquity having the same petrographic classification, physical properties, shape, size, weight, and surface features	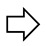	Matching with *Lapis Aequipondus*
3	Deepening the alternative use of *Lapis Aequipondus*	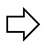	*Lapis Martyrum*
4	Petrographic comparison with compatible (by classification and colour) black serpentinites of Italy	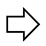	Outlining the most likely provenance area of the *Lapis Aequipondus/Martyrum* and the four black stones of Leopardi’s child home themselves

## Data Availability

Not applicable.

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
