# Peer review of "The Cultural Heritage of “Black Stones” (Lapis Aequipondus/Martyrum) of Leopardi’s Child Home (Recanati, Italy)"

_materials, 2022, doi:10.3390/ma15113828_

Round 1

Reviewer 1 Report

  1. The abstract must be revised with the core novelty of the work.
  2. The introduction section must be elaborated with a few more latest and appropriate references.
  3. Justification for the selection of rocks should be properly elucidated.
  4. The novelty of the current research must be clearly explained in the manuscript. 
  5. The methodology should be given in a flow chart to increase the readability
  6. The discussion on results should be elaborated based on the attained results. However, the current discussion only gives a brief explanation but not a proper justification. Scientific justification of results must be presented.
  7. figures quality must be improved.
  8. English of the manuscript must be improved
  9. conclusions should be revised according to the findings and future scope of the current work also must be stated in this section. 

Reviewer 2 Report

ABSTRACT:

Row 12: these stones instead of these stone

INTRODUCTION

No comments.

MATERIALS AND METHODS

No comments.

RESULTS

Figure 2 shows representative spectra. There are qualitative and quantitative differences between the spectra of the four stones? Please comment or specify this within the text.

It is recommended to analyze a specimen from the same rock native to Tuscany and make a comparison on similar experimental data. Or at least compare it with existing literature data.

DISCUSSION

There is a problem with Figure 3; please check its full position on the page!

Maybe a short comparison on the lithology of every cited stone will be welcomed, by resuming data into a table.

CONCLUSIONS

No comment.

Round 2

Reviewer 1 Report

I feel the manuscript can be accepted in its present form.